# Model-Agnostic Knowledge Guided Correction for Improved Neural Surrogate Rollout

**Bharat Srikishan**[1]*, **Daniel O'Malley**[2], **Mohamed Mehana**[2], **Nicholas Lubbers**[2],
**Nikhil Muralidhar**[1]
[1]Stevens Institute of Technology, [2]Los Alamos National Laboratory
`{bsrikish,nmurali1}@stevens.edu`, `{omalled,mzm,nlubbers}@lanl.gov`

## Abstract

Modeling the evolution of physical systems is critical to many applications in science and engineering. As the evolution of these systems is governed by partial differential equations (PDEs), there are a number of computational simulations which resolve these systems with high accuracy. However, as these simulations incur high computational costs, they are infeasible to be employed for large-scale analysis. A popular alternative to simulators are neural network *surrogates* which are trained in a data-driven manner and are much more computationally efficient. However, these surrogate models suffer from high rollout error when used autoregressively, especially when confronted with training data paucity. Existing work proposes to improve surrogate rollout error by either including physical loss terms directly in the optimization of the model or incorporating computational simulators as 'differentiable layers' in the neural network. Both of these approaches have their challenges, with physical loss functions suffering from slow convergence for stiff PDEs and simulator layers requiring gradients which are not always available, especially in legacy simulators. We propose the Hybrid PDE Predictor with Reinforcement Learning (HyPER) model: a model-agnostic, RL based, cost-aware model which combines a neural surrogate, RL decision model, and a physics simulator (with or without gradients) to reduce surrogate rollout error significantly. In addition to reducing in-distribution rollout error by **47%-78%**, HyPER learns an intelligent policy that is adaptable to changing physical conditions and resistant to noise corruption. Code available at `https://github.com/scailab/HyPER`.

## 1 Introduction

Scientific simulations have been the workhorse enabling novel discoveries across many scientific disciplines. However, executing fine-grained simulations of a scientific process of interest is a costly undertaking requiring large computational resources and long execution times. In the past decade, the advent of low-cost, efficient GPU architectures has enabled the re-emergence of a powerful function approximation paradigm called deep learning (DL). These powerful DL models, with the ability to represent highly non-linear functions can be leveraged as *surrogates* to costly scientific simulations. Recently, the rapid progress of DL has greatly impacted scientific machine learning (SciML) with the development of neural surrogates in numerous application domains. Some highly-impactful applications include protein structure prediction, molecular discovery Schauperl & Denny (2022); Smith et al. (2018) and domains governed by partial differential equations (PDE) Brunton & Kutz (2024); Raissi et al. (2019); Lu et al. (2021b). Neural surrogates have also been successfully employed for modeling fluid dynamics in laminar regimes like modeling blood flow in cardiovascular systems Kissas et al. (2020) and for modeling turbulent Duraisamy et al. (2019) and multi-phase flows Muralidhar et al. (2021); Raj et al. (2023); Siddani et al. (2021).

**Neural Surrogates are Data Hungry**. Although neural surrogates are effective at modeling complex functions, this ability is usually conditioned upon learning from a large trove of representative data. The data-hungry nature of popular neural surrogates like neural operators is well known in

---

*corresponding author

existing work (Li et al., 2020; Lu et al., 2021a; Tripura et al., 2024; Howard et al., 2023; Lu et al., 2022). However, many scientific applications suffer from *data paucity* due to the high cost of the data collection process (i.e., primarily due to high cost of scientific simulations). Hence, neural surrogates employed to model a scientific process of interest, need to address the data paucity bottleneck by learning effectively with a low volume of training data.

**Rollout Errors in Neural Surrogates.** Although computational simulations have been designed for modeling various types of physical systems, those exhibiting transient dynamics are especially challenging to model. Solutions to systems exhibiting transient dynamics are usually obtained by discrete-time evolution of the dynamics. Simulators used to model such systems are invoked autoregressively and thereby encounter numerical instability and error buildup over long estimation horizons. Such error buildup during autoregressive invocation is termed *rollout error*. Effective techniques have been developed to reduce rollout error of computational simulations and increase their numerical stability over long rollouts. Although autoregressive rollout of neural surrogates is also affected by rollout error, solutions to minimize this error buildup have not been widely investigated. Recently, (Margazoglou & Magri, 2023) has inspected the stability of echo-state networks during autoregressive rollout and List et al. (2024); Carey et al. (2024); Lippe et al. (2024) have characterized rollout errors in more general neural surrogates. However, a systematic solution to alleviate rollout error in neural surrogates for modeling transient dynamics is still lacking.

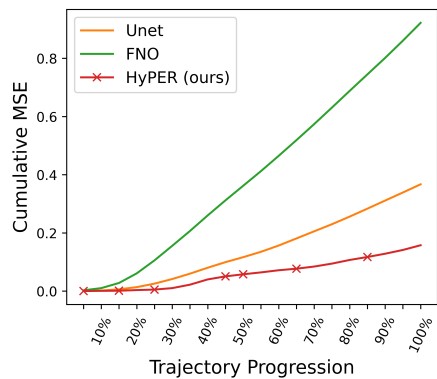

Figure 1: Cumulative MSE depicting *rollout error* for a single trajectory of HyPER vs surrogate only methods. x's mark the timesteps during the trajectory where our RL policy calls the simulator.

**Knowledge-Guided Neural Surrogates.** One popular method of addressing errors due to data paucity in neural surrogates is to leverage knowledge of the theoretical model governing the scientific process. Previous efforts have incorporated domain knowledge (as ODEs, PDEs) while training the DL surrogate to develop knowledge-guided learning pipelines (Raissi et al., 2019; Karpatne et al., 2022; Rackauckas et al., 2020; Gao et al., 2021). Most of these approaches incorporate the PDE governing the system dynamics as soft regularizers while training the neural surrogates. A majority of such approaches exhibit slow convergence and catastrophic failures in challenging, stiff PDE conditions (Krishnapriyan et al., 2021; Wang et al., 2022).

**Hybrid-Modeling** All approaches discussed thus far are so-called *surrogate-only* (SUG) approaches. Here, the computational simulator is employed only as a means of generating data to train the neural surrogate and discarded post the training. SUG approaches employ only the pre-trained neural surrogate during inference. Although SUG provide instantaneous responses relative to computational simulations, they generally have limited generalization ability outside the domain of the training data. An effective complement to SUG approaches are *hybrid-modeling* approaches (Kurz et al., 2022; Karpatne et al., 2022), that jointly resolve a query by incorporating surrogates in conjunction with computational solvers. Otherwise stated, hybrid-modeling pipelines employ a 'solver-in-the-loop' (Um et al., 2020) approach. In addition to neural surrogates, there exist a number of classical hybrid modeling techniques which combine a full order model (FOM) with a reduced order model (ROM) such as proper orthogonal decomposition (Willcox & Peraire, 2002), dynamic model decomposition (Kutz et al., 2016), and multi-scale methods. While these methods are used to accelerate scientific simulations, they are often limited in expressivity, especially in modeling complex non-linear dynamics. Recent hybrid models (Suh et al., 2023) generally have a static coupling between the components in the model and require a static interaction/transition between the FOM and the ROM, while our method proposes an adaptable and learnable interaction between the neural surrogate and the simulator. Our proposed method allows dynamic integration of an FOM (fine-grained simulator) with ROM (neural surrogate) using reinforcement learning.

**Knowledge-Guidance with Hybrid-Modeling.** Hybrid-modeling approaches are inherently knowledge-guided. A majority of the recent hybrid-modeling approaches are based on directly incorporating PDE solvers as additional layers in the deep learning architecture of neural surro-

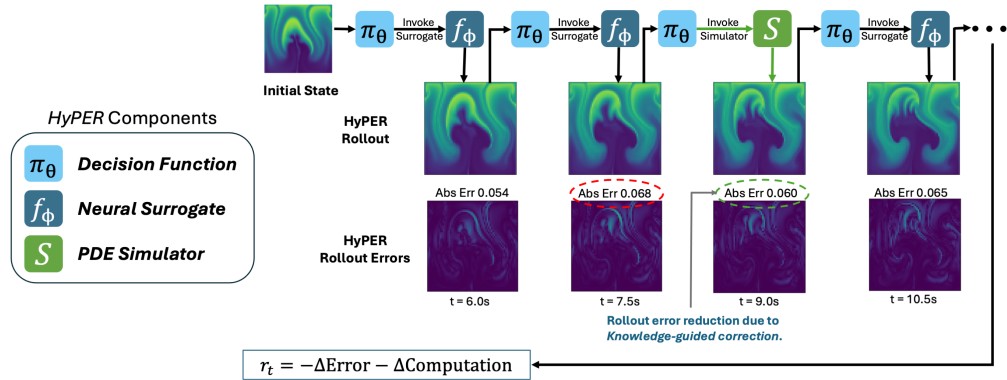

Figure 2: Overview of Hybrid PDE Predictor with RL (HyPER) with example rollout. Here $\pi_\theta$ is the decision model, $f_\phi$ is the surrogate, and $S$ is the simulator. At $t = 9.0$ in the above trajectory, the decision policy invokes the simulator, correcting the trajectory to reduce rollout error. The effect of this knowledge-guided correction can be observed by a reduction in absolute error (dotted green circle) in the figure.

gates (Chen et al., 2018; Belbute-Peres et al., 2020; Donti et al., 2021; Pachalieva et al., 2022). While such approaches address the issues of large errors under data paucity and during rollout, inherent in SUG approaches, they impose the crippling restriction of *differentiability* on the computational solvers to be incorporated as part of the DL pipeline. Most solvers and computational simulations are NOT differentiable out-of-the-box and hence imposing such differentiability constraints drastically curtails the applicability of current hybrid-modeling approaches.

To address the existing challenges with surrogate-only and hybrid modeling approaches, we propose the **Hybrid PDE Predictor with RL** (HyPER) framework. HyPER is a model-agnostic, simulator-agnostic framework that learns to invoke the costly computational simulator (in a cost-aware manner) as *knowledge-guided correction* to alleviate the effects of rollout errors in surrogates trained with low volumes of training data. Fig. 2 depicts the proposed HyPER framework. Our contributions are as follows.

• HyPER is a first of its kind knowledge-guided correction mechanism that incorporates simulators in the loop without the requirement of the simulators to be *differentiable*.

• HyPER is model agnostic (i.e., functions with any neural surrogate, scientific simulator) and trained in a cost-aware manner, to intelligently invoke the simulator to correct the rollout error of the neural surrogates.

• We demonstrate through rigorous experimentation on in-distribution, out-of-distribution and noisy data that HyPER significantly reduces rollout error relative to SUG approaches by comparing with state-of-the-art neural surrogates.

## 2 METHOD

### 2.1 PDE PREDICTION

We aim to solve PDEs involving spatial dimensions $\boldsymbol{x} = [x_1, x_2, \ldots, x_m] \in \mathbb{R}^m$ and scalar time $t \in [0, T]$. These PDEs relate solution function $\boldsymbol{u}(\boldsymbol{x}, t) : \mathbb{R}^m \times [0, T] \to \mathbb{R}^n$ to its partial derivatives over the domain. We assume we have initial conditions $\boldsymbol{u}(\boldsymbol{x}, t = 0)$ and boundary conditions $\boldsymbol{u}_B(\boldsymbol{x} = \boldsymbol{x}_B, t)$ which define the field values at time 0 and at the boundaries of the domain respectively. These time-dependent PDEs can be generally defined as:

$$\frac{\partial \boldsymbol{u}}{\partial t} = \mathcal{F}(\boldsymbol{x}, t, \boldsymbol{u}, \frac{\partial \boldsymbol{u}}{\partial \boldsymbol{x}}, \frac{\partial^2 \boldsymbol{u}}{\partial \boldsymbol{x}^2}, \ldots) \qquad (1)$$

We focus on autoregressive *rollout* of these PDEs over time, which can be defined with an function $g$ taking current state and time as inputs and producing the next state:

$$g(\boldsymbol{u}(\boldsymbol{x}, t), \Delta t) = \boldsymbol{u}(\boldsymbol{x}, t + \Delta t) \qquad (2)$$

Note that the equation above can be applied autoregressively over any number of timesteps to unroll a PDE trajectory. The cumulative error of this autoregressive process is defined as the *rollout error*, which we aim to minimize.

## 2.2 HyPER Components

**Surrogate ML Model**    For the sake of conciseness, we shorten $\boldsymbol{u}(\boldsymbol{x}, t)$ to $\boldsymbol{u}_t$. We begin with a machine learning model $f_\phi(\boldsymbol{u}_t)$, which we denote as the surrogate. This model can be any deep learning model with parameters $\phi$ that predicts next state $\boldsymbol{u}_{t+1}$ given starting state and time $\boldsymbol{u}_t$.

**Simulator**    We also define a PDE simulator $S(\boldsymbol{u}_t)$ which numerically solves the PDE to find the next state $\boldsymbol{u}_{t+1}$. Crucially, this simulator is only required to return the next state without any gradient information during training and inference.

**Decision Model**    Finally, we have a decision model $d_\theta(\boldsymbol{u}_t)$ which takes current state at time $t$ and outputs a next action: either call the surrogate $f_\phi$ or the simulator $S$. In HyPER we implement our decision model as a learned policy $\pi_\theta$ which we train using reinforcement learning (RL).

We formalize our decision model using a Markov Decision Process (MDP) which is a tuple $(\mathbb{S}, \mathbb{A}, P, r)$. Our states $\mathbb{S}$ consist of current state $\boldsymbol{u}$ and current time $t$. Our action space is binary at each timestep and defined as $a : \{0 = \text{call surrogate}, 1 = \text{call simulator}\}$. Our reward function for a trajectory of length $T$ is:

$$R(\boldsymbol{a}) = \sum_{t=0}^{T-1} -\mathcal{L}(f_\phi, S, \boldsymbol{u}_t, \boldsymbol{u}_{t+1}, a_t) + b(f_\phi, S, \boldsymbol{u}_t) - \mathcal{C}(\boldsymbol{a}, \lambda, T) \tag{3}$$

Here $\mathcal{L}$ represents an *error* function, $b$ is a baseline function, and $\mathcal{C}$ is a *cost* function. The baseline function stabilizes the reward values and leads to better policy learning. The baseline function $b$ we use is the mean squared error of randomly calling the simulator the same number of times as our current HyPER policy. Our error function is defined as the mean squared error of either calling the surrogate or simulator according to the policy action:

$$\mathcal{L}(f_\phi, S, \boldsymbol{u}_t, \boldsymbol{u}_{t+1}, a_t) = (\mathcal{G}(\boldsymbol{u}_t) - \boldsymbol{u}_{t+1})^2 \tag{4}$$

$$\mathcal{G}(\boldsymbol{u}_t, a_t) = \begin{cases} f_\phi(\boldsymbol{u}_t) & \text{when } a_t = 0 \\ S(\boldsymbol{u}_t) & \text{when } a_t = 1 \end{cases} \tag{5}$$

$T$ is the length of the trajectory and $\lambda$ is a hyperparameter set by the user to specify what percentage of the trajectory to call the simulator. For example, if $\lambda = 0.5$ then the cost function will penalize the reward if the simulator is not called 50% of the time. Our cost function is defined as the absolute difference between $\lambda$ and the percent of the trajectory that our policy called the simulator:

$$\mathcal{C}(\boldsymbol{a}, \lambda, T) = \left| \frac{\|\boldsymbol{a}\|_1}{T} - \lambda \right| \tag{6}$$

Intuitively, reward function 3 optimizes the RL decision model to minimize mean squared error while calling the simulator for $\lambda$ proportion of the trajectory. To learn a policy $\pi_\theta(a|\boldsymbol{u}_t)$ we use the REINFORCE policy gradient algorithm (Sutton et al., 1999) training with the update:

$$\nabla_\theta J(\theta) = \mathbb{E}_{\pi_\theta} \left[ \sum_{t=0}^{T-1} \nabla_\theta \log \pi_\theta(a_t \mid \boldsymbol{u}_t) \cdot R(\boldsymbol{a}) \right] \tag{7}$$

For the full training algorithm and details of HyPER see Appendix A.1.

## 2.3 Experiments

**2D Navier Stokes Dataset**. We create a 2D incompressible Navier Stokes fluid flow dataset using ΦFlow (Holl & Thuerey, 2024) and follow the example of Gupta & Brandstetter (2022). We use the Navier-Stokes equations in vector velocity form along with an additional scalar field representing particle concentration:

$$\frac{\partial \boldsymbol{v}}{\partial t} = -\boldsymbol{v} \cdot \nabla \boldsymbol{v} + \mu \nabla^2 \boldsymbol{v} - \nabla p + f \tag{8}$$

$$f = \{0, 0.5c\} \tag{9}$$

$$\nabla \cdot \boldsymbol{v} = 0 \tag{10}$$

$$\frac{\partial c}{\partial t} = -\boldsymbol{v} \cdot \nabla c \tag{11}$$

Eq. 8 comprises a convection term $-\boldsymbol{v} \cdot \nabla \boldsymbol{v}$, diffusion term $\mu \nabla^2 \boldsymbol{v}$ where $\mu$ indicates kinematic viscosity, a pressure term $\nabla p$, and external force term $f$. Eq. 9 defines the buoyant force which is applied in the y-direction. The velocity divergence term Eq. 10 enforces conservation of mass. Eq. 11 models the particle concentration field $c$ that is advected by the velocity vector field $\boldsymbol{v}$. Note that $c$ influences $\boldsymbol{v}$ through force $f$ and $\boldsymbol{v}$ affects $c$ through Eq. 11, creating complex dynamics between fields. For our experiments, we generate 1,000 trajectories of 20 timesteps with $\Delta t = 1.5s$ at a grid size of 64x64, and diffusion coefficient $\mu = 0.01$. We set our velocity boundary condition to $\boldsymbol{v} = 0$ (Dirichlet) and our concentration boundary condition to $\partial c / \partial \boldsymbol{x} = 0$ (Neumann). While we simulate all the fields above, our experiments focus on predicting the particle concentration field $c$. We split our 1000 trajectories into 3 sets: 400 for surrogate training, 400 for RL training, and 200 for testing. Our RL model selects 4 actions at a time, however HyPER puts no restrictions on action selection.

**Subsurface Flow Dataset**. To evaluate HyPER's ability to work with different PDEs and problem scales, we generate a subsurface flow dataset using the Julia-based DPFEHM (Pachalieva et al., 2022) simulator. We use DPFEHM to generate a 2D dataset of subsurface fluid flows modeled by the Richards equation:.

$$\frac{\partial \theta}{\partial t} = \nabla \cdot \mathbf{K}(h)(\nabla h + \nabla z) - T^{-1} \tag{12}$$

Eq. 12 models fluid flow underground in an unsaturated medium. $\theta$ represents the volumetric fluid content, $\mathbf{K}(h)$ is the unsaturated hydraulic conductivity, $h$ is the pressure, $\nabla z$ is the geodetic head gradient, and $T^{-1}$ is the fluid sink term. We generate 500 trajectories of 100 timesteps each with a grid size of 50x50 and timestep size of 10 seconds. Out of the 500 trajectories, 200 are used for surrogate training, 200 are used for RL training, and 100 are used for testing. We use Dirchlet boundary conditions (0) for volumetric flux on all sides of our domain but inject fluid at the top center of the domain at a rate of $0.01 m/s$. In our experiments, we predict the pressure field $h$.

## 3 RESULTS AND DISCUSSION

In this section, we investigate the effectiveness of HyPER to model transient PDE systems efficiently and with minimal rollout error compared to surrogate-only (SUG) rollout. We compare with state-of-the-art neural surrogates including UNet and Fourier Neural Operator (FNO), MPP, and PDE-Refiner. Specifically, we investigate the effectiveness of HyPER rollouts under changing PDE dynamics and in noisy data settings. Further, we also demonstrate the surrogate-agnostic and simulator-agnostic nature of HyPER. Our experiments seek to answer the following research questions:

• **RQ1:** How effective are HyPER rollouts compared to SUG rollouts for transient PDE systems?

• **RQ2:** Are HyPER rollouts effective under changed physical conditions?

• **RQ3:** Are HyPER rollouts effective under noisy data conditions?

• **RQ4:** How crucial is the intelligent decision model for effective HyPER rollouts?

• **RQ5:** What is the error/efficiency trade-off between SUG, HyPER, and simulator-only paradigms?

### 3.1 RQ1: HYPER ROLLOUTS VS SUG

We begin by comparing HyPER to six SUG baselines: *UNet*, *FNO*, *UNet-Multistep*, *MPP-ZS*, *MPP*, and *PDE-Refiner*. All baselines except UNet-Multistep are trained in a one-step-ahead manner i.e., given the current state of the resolved field they are trained to predict the next state. Mean squared error (MSE) loss is used to train all surrogates over the full resolution trajectory. Note that all these baselines except MPP-ZS are trained with the same dataset that HyPER is trained with. For full training details see Appendix A.1 and for baseline details see Appendix A.6.

The UNet model is based on the modern UNet architecture in PDEArena (Gupta & Brandstetter, 2022) while the FNO model is built with the *neuraloperator* library (Kovachki et al., 2021; Li et al.,

Table 1: 2D Navier-Stokes results with best results in bold. Notice that HyPER rollout incurs significantly lower rollout (cumulative) error compared to SUG models.

| 2D Navier-Stokes | UNet | FNO | UNet-Multistep | MPP-ZS | MPP | PDE-Refiner | HyPER |
|---|---|---|---|---|---|---|---|
| Final MSE | 0.019 | 0.053 | 0.016 | 0.22 | 0.036 | 0.024 | **0.011** |
| Cumulative MSE | 0.312 | 0.882 | 0.311 | 4.495 | 0.747 | 0.405 | **0.164** |

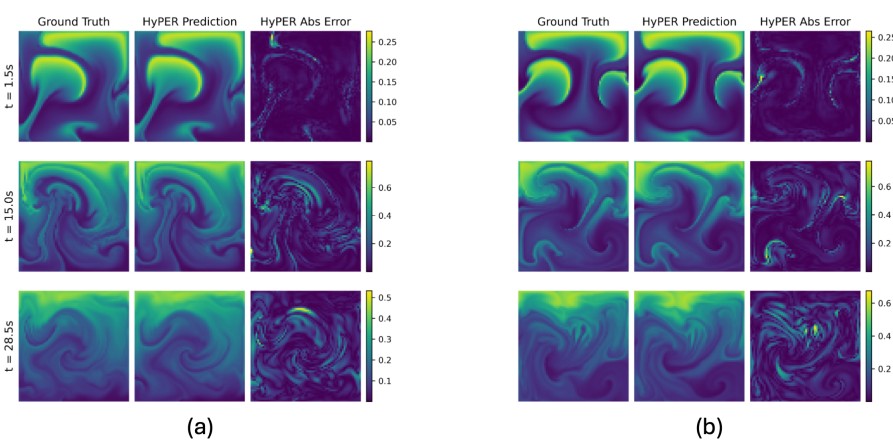

Figure 3: Predictions and absolute error snapshots of HyPER rollout for two distinct trajectories.

2020). UNet-Multistep is trained to specialize for rollout trajectory prediction, i.e. it minimizes MSE for 20-step autoregressive prediction. MPP-ZS (zero-shot) and MPP (McCabe et al., 2023) are large transformer based multi-task models which have been pretrained on a variety of fluid prediction tasks. MPP-ZS is the off-the-shelf pretrained model, while MPP is finetuned on our data. PDE-Refiner (Lippe et al., 2024) is a modern diffusion based model which both predicts the next timestep while adding a multi-step noising/denoising process and has been shown to reduce rollout error. HyPER utilizes a UNet model that is pretrained on 400 trajectories while the RL model of HyPER is trained on a separate set of 400 trajectories with $\lambda = 0.3$. All SUG baselines are trained using 800 trajectories (same as HyPER RL plus surrogate dataset) for 200 epochs. Table 1 shows the MSE of the final trajectory state (Final MSE) as well as the aggregated MSE (Cumulative MSE) over all trajectory states in a rollout. Both metrics are calculated as an average across 200 test trajectories for both SUG and HyPER rollouts. HyPER outperforms all baselines significantly, with an average improvement in cumulative rollout error of **68.30%**. While UNet-Multistep seems to be the best performing SUG method, because it trains across the full trajectory it learns a smoothed prediction that has poor high frequency detail (suffering from spectral bias) in comparison to UNet (see A.4 for examples). By incorporating the simulator, HyPER yields significantly lower cumulative rollout error while only invoking the simulator during $\sim 30\%$ of the trajectory for knowledge-guided correction of surrogate rollout.

In Figure 1 we plot the rollout MSE performance of HyPER versus the surrogate only (SUG) methods for a single sample trajectory. Note that HyPER outperforms both SUG models by a large margin while only invoking the simulator six times in the 20-step trajectory. Also see Figures 3(a) and 3(b) which depict qualitative snapshots from two distinct HyPER rollouts.

## 3.2 RQ2: HyPER ROLLOUTS WITH CHANGING PHYSICAL CONDITIONS

Pretrained neural surrogates are often confronted with modeling contexts with similar PDE dynamics but varied initial / boundary conditions. Effective rollouts under changing physical conditions is hence a crucial requirement for the effective use of neural surrogates in computational science. To this end, we inspect rollout errors of SUG and HyPER, under changing boundary conditions. Specifically, to test the adaptability of HyPER, we investigate HyPER rollouts with changing boundary conditions in our Navier-Stokes experiment. We generate a separate Navier-Stokes dataset which follows the same initial conditions of our previous experiment, but comprises a different velocity

boundary condition at the top boundary of the domain. Specifically, the velocity at the top boundary of the domain is increased from 0.0 to 0.5 m/s (imposing an intermittent external forcing effect causing the fluid to escape from the top of the domain) for four intermediate time-steps of the trajectory (timesteps 12-16). Following this, the boundary condition is reverted back to 0.0 for the rest of the trajectory. This results in the fluid escaping out of the top during these time steps changing the PDE dynamics significantly.

Table 2: Results depict rollout error for 2D Navier-Stokes with changing boundary conditions. Notice that HyPER accumulates lower rollout error compared to SUG approaches.

| Changing Boundary | UNet-P | FNO-P | UNet | FNO | HyPER |
|---|---|---|---|---|---|
| Final MSE | 0.362 | 0.412 | **0.014** | 0.027 | 0.027 |
| Cumulative MSE | 1.84 | 2.538 | 0.92 | 0.969 | **0.527** |

In this case, surrogates UNet-P and FNO-P are not re-trained with the changed boundary condition and hence show poor performance accumulating significant rollout error. UNet and FNO are trained on the changing boundary data (the same dataset as HyPER), but still suffer from poor rollout error. We train our HyPER RL policy with a pretrained surrogate (not trained on to the changed boundary setting) and a simulator (fully aware of the changed boundary condition). The RL policy of HyPER is trained with 400 of these changed trajectories to learn the optimal policy of (frugally) invoking the simulator to apply knowledge-guided correction to reduce surrogate rollout errors. As shown in Table 2, HyPER reduces cumulative error in this scenario by **71.36%** and **79.20%** relative to UNet-P and FNO-P rollout errors respectively. Even when compared to surrogates trained on the changing boundary data UNet and FNO, HyPER reduces rollout error by **42.72%** and **45.61%** respectively. In Figure 4a we demonstrate that HyPER has much lower rollout error than the UNet-P and FNO-P models. Figure 4b shows sample qualitative predictions of HyPER and SUG rollouts under the changing boundary scenario of interest, to further reinforce our point. As illustrated, our model prediction is much closer to the ground truth after the boundary condition has changed and fluid has escaped the box owing to the appropriately invoked knowledge-guided correction by the learned RL policy in response to increasing surrogate rollout error. Fig. 4a shows the significantly lower rollout error for HyPER rollout relative to UNet-P and FNO-P models.

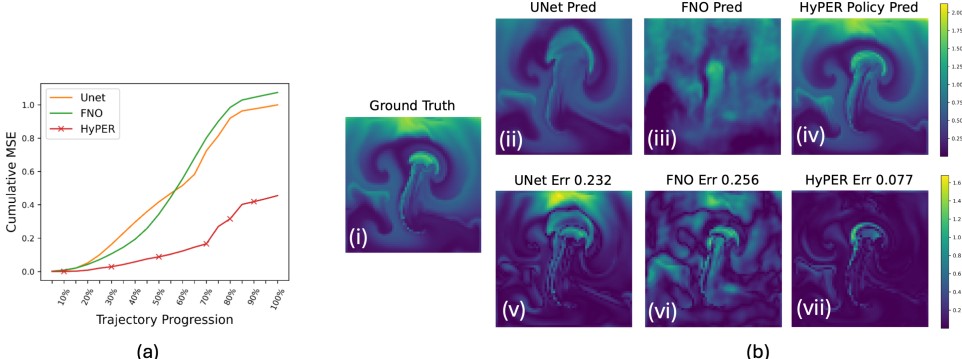

(a)                                                                 (b)

Figure 4: Predictions and absolute error of HyPER vs UNet and FNO for a single trajectory and timestep. Fig. 4a shows the rollout error accumulation over the trajectory with 'x' marking the times the simulator is called. Fig. 4b shows the resolved system state for a the same trajectory at a single timestep. Fig. 4b(i) shows the ground truth field while Fig. 4b(ii)-(iv) indicate the result of UNet, FNO and HyPER rollouts respectively. Fig. 4b(v)-(vii) depict the corresponding absolute errors. We notice that only HyPER rollouts capture the correct characteristics relative to the ground truth owing to the knowledge-guided correction while SUG models are unable to faithfully resolve the trajectory under changing physical conditions.

### 3.3 RQ3: HyPER ROLLOUTS WITH NOISY DATA

Neural surrogates, in addition to being data hungry and accumulating rollout error, have also been known to exhibit catastrophic failures in challenging PDE conditions (e.g., stiff PDEs) Krishnapriyan et al. (2021). As neural surrogates are a crucial part of HyPER rollouts, it is imperative

Table 3: Results depicting rollout error for the 2D-Navier-Stokes experiment, with random Gaussian noise added to inputs, at timesteps 12-16. $\sigma^2$ is the *scale* of the noise. The percentage reduction of cumulative MSE rollout error by HyPER, over the best performing SUG model is in parentheses.

| Experiment | Cumulative MSE | | | | |
|---|---|---|---|---|---|
| | UNet-P | FNO-P | UNet | FNO | HyPER |
| Unimodal $\sigma^2 = 1.0$ | 2.931 | 2.523 | 1.61 | 1.713 | **1.291** (19.81%) |
| Unimodal $\sigma^2 = 0.75$ | 1.746 | 1.916 | 1.013 | 1.291 | **0.8** (21.03%) |
| Unimodal $\sigma^2 = 0.5$ | 1.009 | 1.487 | 0.642 | 1.137 | **0.49** (23.68%) |
| Unimodal $\sigma^2 = 0.25$ | 0.612 | 1.246 | 0.519 | 0.862 | **0.26** (49.90%) |

to investigate whether HyPER is capable of *adapting* to such failures by invoking the knowledge-guided correction (i.e., the simulator) to minimize the propagation (and accumulation) of such local failures over the remaining trajectory rollout. Further, another crucial property to investigate is the robustness of the intelligent decision mechanism in HyPER, to noisy, low-quality surrogate predictions. To jointly investigate both goals, we consider a simple (contrived) context with noisy inputs supplied to the HyPER RL policy. This additive noise, injected at specific steps to *corrupt* the surrogate output in the trajectory, serves to mimic low-quality surrogate predictions. Hence, experiments with such noisy inputs help characterize the ability of HyPER to adapt to sudden local changes during rollout (like catastrophic surrogate failure) in addition to demonstrating its ability for robust decision-making under noisy data conditions.

To carry out this investigation, we add random Gaussian noise with mean 0 at four separate variance ($\sigma^2$) scales at fixed timesteps of our trajectory. The UNet-P and FNO-P surrogate models are never trained with this noisy data and therefore perform poorly when encountering it. The UNet and FNO models are trained with the noisy data and show improvement. We train the RL policy of HyPER with a small set (400) of these noisy trajectories while keeping our surrogate model static (using the UNet-P model). We test two different noise corruption scenarios added to a 20 step trajectory rollout. (i) unimodal noise: a case where noise is added at timesteps [12-16] and (ii) bimodal noise: a more sophisticated case where noise is added at two different time windows of [2-4] and [15-16]. The unimodal noise results are presented in Table 3, where we see that HyPER rollouts outperform state-of-the-art SUG approaches. In this case we notice a reduction in cumulative MSE by **19.81**%-**49.90**% across the four noise scales. Note that HyPER achieves this significant reduction in rollout error while using the un-specialized UNet-P surrogate, demonstrating that HyPER succeeds in adapting pretrained models to noisy data.

### 3.4 RQ4: INVESTIGATING HyPER ROLLOUTS VS. RANDOM POLICY ROLLOUTS

To evaluate whether HyPER learns an effective RL policy, we compare it to a Random Policy baseline. This baseline is designed to invoke the simulator the same number of times as the HyPER RL policy for a particular rollout, but with uniform random probability over each timestep. By comparing the HyPER policy rollout to a Random Policy rollout with the same budget, we show that HyPER learns a superior performing policy in our experimental scenarios.

Table 4: HyPER versus a random policy which calls the simulator the same number of times.

| Experiment | Final MSE | | Cumulative MSE | | Cumulative Wins % |
|---|---|---|---|---|---|
| | Random Policy | HyPER | Random Policy | HyPER | HyPER |
| Noise Free | **0.011** | **0.011** | 0.186 | **0.164** | 57.00% |
| Bimodal $\sigma^2 = 1.0$ | 0.121 | **0.07** | 2.443 | **1.706** | 73.00% |
| Bimodal $\sigma^2 = 0.75$ | 0.055 | **0.025** | 1.305 | **0.739** | 84.50% |
| Bimodal $\sigma^2 = 0.5$ | 0.037 | **0.027** | 0.891 | **0.683** | 67.00% |
| Bimodal $\sigma^2 = 0.25$ | 0.022 | **0.019** | 0.495 | **0.424** | 61.00% |
| Changing Boundary | 0.149 | **0.027** | 0.865 | **0.527** | 80.50% |

We summarize these results in Table 4, which also shows the percentage of trajectories over which HyPER reduces MSE compared to Random Policy (Cumulative Wins %). A 'win' is characterized by a HyPER rollout that yields a lower cumulative MSE than the corresponding Random Policy rollout. In the 'Noise Free' trajectories, HyPER only wins for 57% of the trajectories because the MSEs at each timestep in the Noise Free case have low variance, so a uniform Random Policy performs fairly well. However, in the bimodal noise scales (with fast and large rollout error accumulation), precise invocation of the simulator for knowledge-guided correction is imperative to prevent error accumulation. Hence, we see HyPER owing to its intelligent (RL-based) decision policy, has a higher percentage of wins ($\approx 70\%$) and significantly lower cumulative MSE at higher noise scales. We also see the strength of HyPER's learned policy when considering the changing boundary condition experiment which has a 80.50% win rate over the Random Policy and reduces cumulative MSE by 39.08%. This result demonstrates the effectiveness of HyPER in realistic physical scenarios and noisy conditions.

## 3.5 RQ5: COST VERSUS ACCURACY TRADE-OFF

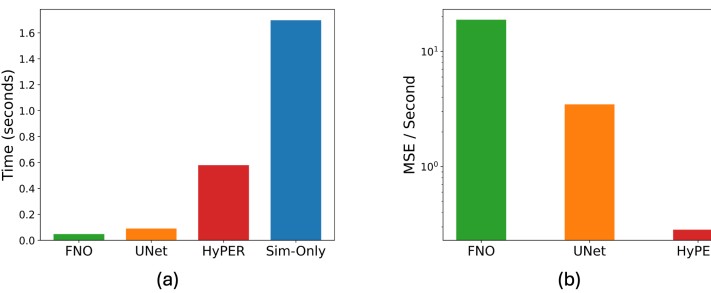

Figure 5: Figure 5(a) shows the average time of PDE prediction of a full trajectory for each method. Figure 5(b) illustrates the error per unit time (*lower is better*) for each method. We do not show the Sim-Only case here as we assume error is effectively zero for the simulator.

A natural question is how much of a cost we pay in wall clock time when utilizing HyPER compared to baselines? We evaluate this by measuring the average time of each method on our 200 test trajectories and show results in Figure 5(a). While the wall clock time of SUG methods is lower than HyPER, our method has much lower rollout error, which we illustrate in Figure 5(b). Here we show the error per unit time (*lower is better*) and we see that HyPER has lower rollout error per second by a large margin in comparison to the baselines (note this plot is on a log scale). While calling the simulator incurs a time cost, the user can specify the $\lambda$ parameter to adjust to their requirements allowing HyPER to be flexible in varying settings.

## 3.6 SURROGATE-AGNOSTIC AND SIMULATOR-AGNOSTIC DESIGN OF HYPER

To demonstrate the model-agnostic capability of HyPER, we train it on the subsurface flow task (dataset details in Sec. 2.3), comprising longer (i.e., 100 step) trajectories. In this case, we train HyPER using the Julia simulator DPFEHM, which simulates underground fluid flow in porous media. As we see in Table 5, HyPER outperforms both SUG baselines by a cumulative MSE per timestep

Table 5: 2D Subsurface flow experiment results for a 100 timestep trajectory.

| Experiment | Cumulative MSE per timestep | | |
| --- | --- | --- | --- |
| | UNet-P | FNO-P | HyPER |
| Subsurface | 4.345 | 10.938 | **0.271** |

reduction of 93.76% and 97.52%. The integration of a distinct Julia-based simulator in a scenario with larger physical scales and times, shows that HyPER is surrogate and simulator agnostic and can perform well and reduce rollout error in multiple problem settings.

## 4 Related Work

**SUG** approaches circumvent the use of computational simulations during inference and only employ simulations to generate training data. The U-Net (Ronneberger et al., 2015) model is a popular approach owing to its ability to capture spatial and temporal dynamics at multiple scales. This model has recently (Gupta & Brandstetter, 2022) demonstrated state of the art performance on various fluid dynamics benchmarks. Operator learning (Kovachki et al., 2021) approaches that learn function families of PDEs rather than single PDE instances have also been investigated to be effective neural surrogates. Two notable operator learning models are the deep operator network (Lu et al., 2021a) and the Fourier neural operator (FNO) (Li et al., 2020) models. Multiple investigations have been carried out employing operator learning techniques including extending them to multi-resolution (Howard et al., 2023; Lu et al., 2022) settings. Recently Takamoto et al. (2022) have also demonstrated that FNOs yield state-of-the-art results on benchmark tasks.

**Knowledge-guided SUG** approaches like the popular Physics-informed neural network (PINN) (Raissi et al., 2019; Jagtap & Karniadakis, 2020; Cuomo et al., 2022) have also been effectively employed for improved generalization. A related paradigm of Universal Differential Equations (Rackauckas et al., 2020) utilizes data-driven surrogates to estimate focused sub-components of governing equations for better process modeling. Such approaches assume end-to-end gradient based training in a physics-informed manner and are known to converge slowly and to trivial solutions under data paucity and stiff PDE conditions Krishnapriyan et al. (2021) owing to catastrophic gradient imbalance Wang et al. (2021) between data-driven and physics-guided loss terms.

**SUG for Transient PDE Dynamics.** All SUG approaches struggle to model transient PDE systems in an autoregressive manner and incur rollout error (Carey et al., 2024). In the work of Lippe et al. (2024), it is demonstrated how the spectral bias of traditional neural surrogates leads to significant error accumulation and they propose an initial *refinement* solution inspired by the process in diffusion modeling. Separately, List et al. (2024) have conducted a characterization of SUG rollout error and comment about the significant improvement obtainable by incorporating simulators in-the-loop.

**Hybrid-modeling for Transient PDE Dynamics.** Hybrid-modeling approaches retain the simulation and resolve each query employing the neural surrogate and the simulator 'in-the-loop'. One recent example is Zhang et al. (2022) which selects between a simulator and a surrogate (in a pre-defined rule-based maner) to resolve a PDE trajectory. In their work, the simulator and surrogate are each invoked a pre-fixed number of times. In contract, HyPER learns a dynamic policy, capable of adapting to scenarios such as changing boundary conditions. Another major drawback with many such approaches (Chen et al., 2018; Belbute-Peres et al., 2020; Um et al., 2020; Donti et al., 2021; Pachalieva et al., 2022) is the requirement of simulators to be *differentiable* as they are mostly employed as additional layers in the neural network architecture, to be trained end-to-end with the neural surrogates.

## 5 Conclusion and Future Work

This work presents a first of its kind knowledge-guided correction mechanism to reduce rollout errors in neural surrogates that model transient PDE systems. Our proposed method Hybrid PDE Predictor with RL (HyPER) learns a reinforcement learning based cost-aware control policy to parsimoniously invoke (costly) simulation steps to *correct* erroneous surrogate predictions. In contrast to existing approaches that employ simulators 'in-the-loop' with neural surrogates, HyPER does not impose any differentiability restrictions on the computational simulations. Further, HyPER is surrogate and simulator agnostic and is applicable to any neural surrogate and off-the-shelf simulator capable of resolving transient PDE systems.

We have demonstrated the effectiveness of our proposed HyPER model in traditional in-distribution rollouts, under changing physical conditions and under noisy data conditions. Overall HyPER yields significant improvements of cumulative rollout error over state-of-the-art surrogate-only approaches with an average **68.30**% improvement for in-distribution rollouts, **75.28**% improvement for rollouts under changing physical conditions and **28.61**% improvement for rollouts under noisy data conditions. In the future, we will investigate more sophisticated actor-critic based RL policies based to further improve the sample efficiency of HyPER. We will also explore extensions of HyPER to more challenging problems in multi-physics contexts as well as multi-phase flows.

ACKNOWLEDGMENTS

DO gratefully acknowledges support from the Department of Energy, Office of Science, Office of Basic Energy Sciences, Geoscience Research program under Award Number LANLECA1.

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

# A APPENDIX

## A.1 HyPER TRAINING PROCEDURE

Below we detail the RL training procedure of HyPER.

---

**Algorithm 1** HyPER Training Algorithm

---

**Require:** Dataset: $\mathcal{D}$, Pretrained Surrogate Model: $f_\phi$, Simulator: $S$, Decision Model: $d_\theta$,
Trajectory length: $\tau$, Simulator proportion hyperparameter: $\lambda$, Learning rate: $\eta$,
Error function (MSE in our case): $\ell$, Cost function: $c$

1: **for** $\boldsymbol{u}, \boldsymbol{z} \in \mathcal{D}$ **do**     ▷ For every trajectory in dataset, get field $\boldsymbol{u}$ and conditional features $\boldsymbol{z}$
2:     $k \leftarrow 0$     ▷ Initialize number of simulator calls
3:     $R_d \leftarrow [\,]$     ▷ Initialize decision model rewards list
4:     $R_b \leftarrow [\,]$     ▷ Initialize baseline rewards list
5:     $L \leftarrow [\,]$     ▷ List to store log probabilities of each action
6:     $\hat{\boldsymbol{u}}(\boldsymbol{x}, -1) \leftarrow 0$     ▷ Set initial field prediction to 0
7:     **for** $t \in [0, \tau]$ **do**     ▷ For every time-step in trajectory
8:        $p \leftarrow d_\theta(\hat{\boldsymbol{u}}(\boldsymbol{x}, t-1), \boldsymbol{z}, t)$     ▷ Get RL model probabilities for next action
9:        $a \sim \text{Bernoulli}(p)$     ▷ Sample next action
10:        $L \mathrel{+}= a \log(p) + (1-a) \log(1-p)$     ▷ Store log probability of action
11:        **if** $a = 0$ **then**
12:           $\hat{\boldsymbol{u}}(\boldsymbol{x}, t) \leftarrow f_\phi(\hat{\boldsymbol{u}}(\boldsymbol{x}, t-1), t)$     ▷ Call surrogate for next step prediction
13:        **else if** $a = 1$ **then**
14:           $\hat{\boldsymbol{u}}(\boldsymbol{x}, t) \leftarrow S(\hat{\boldsymbol{u}}(\boldsymbol{x}, t-1), t)$     ▷ Call simulator for next step prediction
15:           $k \mathrel{+}= 1$     ▷ Track number of times simulator called
16:        **end if**
17:        $R_d \mathrel{+}= -\ell(\hat{\boldsymbol{u}}(\boldsymbol{x}, t), \boldsymbol{u}(\boldsymbol{x}, t)) - c(\lambda, k, \tau)$     ▷ Store policy reward based on MSE and cost function
18:     **end for**
19:     $I \sim \text{UniformWithoutReplacement}([0, \tau], k)$     ▷ Sample $k$ times between $[0, \tau]$ without replacement, this list will contain the time-steps at which the random baseline will call the simulator
20:     **for** $t \in [0, \tau]$ **do**     ▷ Run random baseline for trajectory
21:        **if** $t \notin I$ **then**
22:           $\hat{\boldsymbol{u}}(\boldsymbol{x}, t) \leftarrow f_\phi(\hat{\boldsymbol{u}}(\boldsymbol{x}, t-1), t)$     ▷ Call surrogate for next step prediction
23:        **else if** $t \in I$ **then**
24:           $\hat{\boldsymbol{u}}(\boldsymbol{x}, t) \leftarrow S(\hat{\boldsymbol{u}}(\boldsymbol{x}, t-1), t)$     ▷ Call simulator for next step prediction
25:        **end if**
26:        $R_b \mathrel{+}= -\ell(\hat{\boldsymbol{u}}(\boldsymbol{x}, t), \boldsymbol{u}(\boldsymbol{x}, t)) - c(\lambda, k, \tau)$     ▷ Store baseline reward using MSE and cost function
27:     **end for**
28:     $\nabla_\theta J \leftarrow -\nabla_\theta L \cdot \text{StopGradient}(R_d - R_b)$     ▷ Calculate policy gradient, this is done element-wise and then summed
29:     $\theta \leftarrow \theta - \eta \nabla_\theta J$     ▷ Update RL decision model parameters
30: **end for**

---

## A.2 RQ3: HyPER ROLLOUTS WITH NOISY DATA: BIMODAL NOISE EXPERIMENTS

Table 6 depicts the performance of HyPER in the bimodal noise case. We notice results similar to the unimodal noise case, with a reduction in cumulative MSE error of **41.46%-65.24%**. We believe this improved performance in a more complex noise distribution is a result of our RL policy learning the more complex error distribution while the SUG methods accumulate larger error over two different noise windows.

Table 6: Results depicting rollout error for the 2D-Navier-Stokes experiment with random Gaussian noise added at timesteps 2-4, 15-16. $\sigma^2$ is the *scale* of the noise. The percentage reduction of cumulative MSE rollout error by HyPER, over the best performing SUG model is in parentheses.

| Experiment | Final MSE | | | Cumulative MSE | | |
|---|---|---|---|---|---|---|
| | UNet-P | FNO-P | HyPER | UNet-P | FNO-P | HyPER |
| Bimodal $\sigma^2 = 1.0$ | 0.844 | 0.097 | **0.07** | 6.648 | 2.914 | **1.706** (41.46%) |
| Bimodal $\sigma^2 = 0.75$ | 0.25 | 0.079 | **0.025** | 3.436 | 2.126 | **0.739** (65.24%) |
| Bimodal $\sigma^2 = 0.5$ | 0.087 | 0.07 | **0.027** | 1.942 | 1.576 | **0.683** (56.67%) |
| Bimodal $\sigma^2 = 0.25$ | 0.046 | 0.068 | **0.019** | 1.095 | 1.266 | **0.424** (61.28%) |

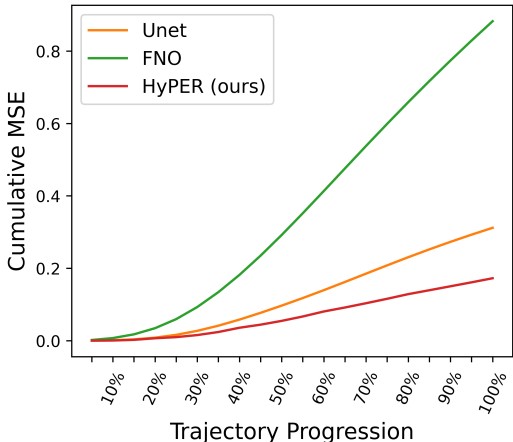

Figure 6: Average Cumulative MSE over all test trajectories of HyPER vs. UNet and FNO.

## A.3 HYPER AVERAGE PERFORMANCE

Figure 6 shows the average rollout error of HyPER versus UNet and FNO methods for all 200 Navier-Stokes test trajectories. This demonstrates that HyPER is effective in reducing rollout error significantly over a large set of unseen test trajectories.

## A.4 UNET-MULTISTEP PREDICTION QUALITY

In Figure 7 below we show five different sample predictions of UNet versus UNet-Multistep. Our investigations revealed that in comparison to UNet, all UNet-Multistep predictions look "smoothed". Essentially we found that while UNet-Multistep had reasonably low MSE, it failed to capture high frequency detail and suffered from spectral bias. Thus, UNet was chosen as our surrogate model in HyPER across all experiments because it showed comparable cumulative MSE to UNet-Multistep (Table 1).

## A.5 HYPER VERSUS UNET PREDICTION

We contrast the predictions and errors of a HyPER rollout and UNet rollout in Figure 8. Notice that at $t = \{16.5s, 21.0s\}$, UNet suffers from spectral bias and has high error on the plume edges where high frequency detail is required. We see that HyPER successfully mitigates this spectral bias error by incorporating the simulator, thereby significantly reducing rollout (cumulative) error over the full trajectory.

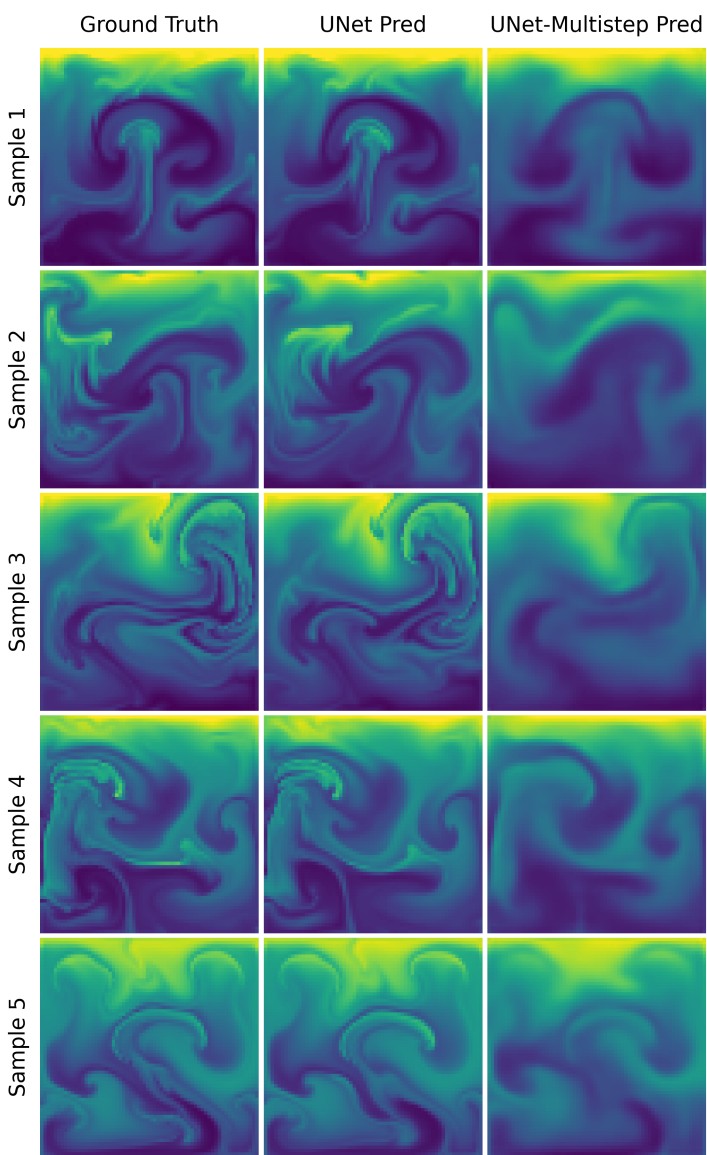

Figure 7: Sample predictions of UNet vs UNet-Multistep on five different noise free trajectories. Note that UNet-Multistep predictions suffer from spectral bias and have poor high frequency detail.

## A.6    MODEL ARCHITECTURES AND PARAMETERS

### A.6.1    MODEL TRAINING DETAILS

We build and train all our neural surrogate models using the PyTorch library on a single Nvidia RTX A6000 GPU. The ΦFlow simulator used in the Navier-Stokes experiment runs on the same GPU. The DPFEHM Julia simulator runs using multi-threading on an Intel(R) Xeon(R) Platinum 8358 CPU @ 2.60GHz.

The UNet, FNO, UNet-Multistep, MPP, and PDE-Refiner baselines are trained with same number of samples as HyPER for the sake of fair comparison. The UNet-P and FNO-P baseline models are trained with a smaller dataset of 400 trajectories which are separate from the 400 trajectories HyPER's RL policy is trained with. All baselines are trained for 200 epochs.

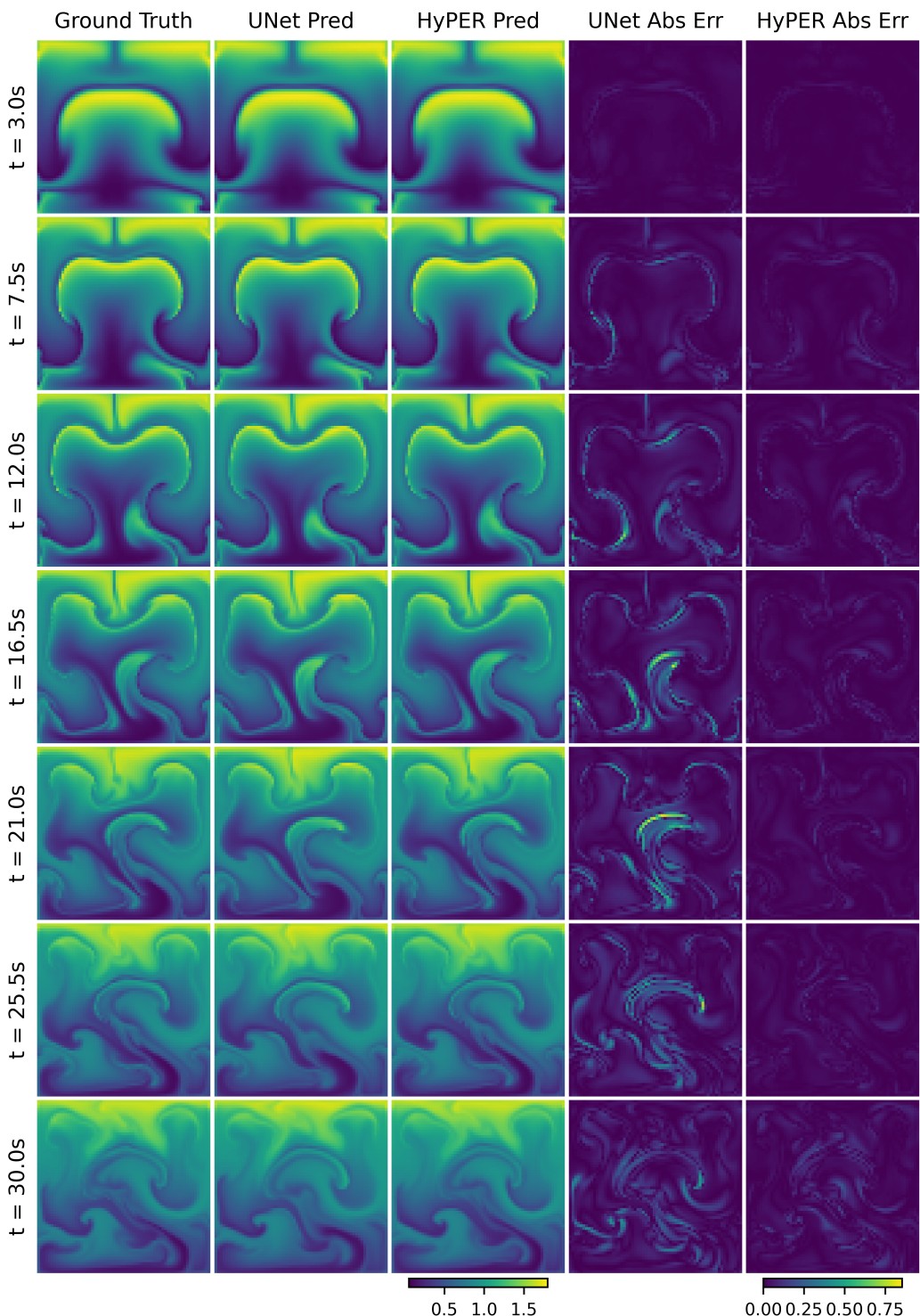

Figure 8: Sample trajectory predictions and absolute errors of HyPER vs UNet. At various timesteps including $t = 21.0s$, we see that UNet has high error on the edges of the plume where high frequency behavior is present while HyPER successfully reduces this error.

We only train the HyPER RL ResNet model for 30 epochs to demonstrate that HyPER does not require extensive training to outperform the baselines.

Table 7: Model training times and datasets for SUG methods and HyPER.

| Model | Training Time (Hours) | Epochs | Dataset Size (# Trajectories) |
|---|---|---|---|
| UNet-P | 1.46 | 200 | 400 |
| FNO-P | 0.55 | 200 | 400 |
| UNet | 3.29 | 200 | 800 |
| FNO | 2.02 | 200. | 800 |
| UNet-Multistep | 4.10 | 200 | 800 |
| MPP | 9.58 | 200 | 800 |
| PDE-Refiner | 3.66 | 200 | 800 |
| HyPER | 5.21 | 30 | 400 |

### A.6.2 MODEL PARAMETERS

Our UNet model is built on top of the PDEArena modern UNet architecture with wide residual blocks and training parameters in Table 8.

Table 8: UNet parameters.

| Parameter | Value |
|---|---|
| Model Size (# parameters) | 12,295,233 |
| Hidden Channels | 64 |
| Activation Function | GELU |
| Channel Multipliers | [1, 2, 2] |
| Num Residual Blocks Per Channel | 2 |
| Sinusoidal Time Embedding | Yes |
| Learning Rate | 1e-4 |
| Optimizer | Adam |
| Loss Function | MSE |
| Epochs | 200 |

Our FNO model is built using the neuraloperator library and is constructed to have a similar number of parameters as our UNet for a fair comparison. See Table 9 for details.

Table 9: FNO parameters.

| Parameter | Value |
|---|---|
| Model Size (# parameters) | 12,437,057 |
| Number of Fourier Modes | 27 |
| Hidden Channels | 64 |
| Lifting Channels | 256 |
| Projection Channels | 256 |
| Learning Rate | 1e-5 |
| Optimizer | Adam |
| Loss Function | MSE |
| Epochs | 200 |

The HyPER RL model is a lightly modified version of the ResNet34 model from the Torchvision library. See Table 10 for parameters.

The Multiple Physics Pretrained model (MPP) is adapted from MPP github. Note that we train this model for 200 epochs while UNet and FNO are only trained for 50. See Table 11 for details.

Table 10: RL ResNet parameters.

| Parameter | Value |
|---|---|
| Model Size (# parameters) | 11,751,300 |
| Number of Layers | [3, 4, 6, 3] |
| Hidden Channels | 64 |
| Sinusoidal. Time Embedding | Yes |
| Activation Function | ReLU |
| Learning Rate | 1e-5 |
| Optimizer | Adam |
| Reward Function | Equation 3 |
| Epochs | 30 |

Table 11: Multiple Physics Pretraining parameters.

| Parameter | Value |
|---|---|
| Model Size (# parameters) | 28,979,436 |
| Patch Size | 16x16 |
| Embedding Dimension | 384 |
| Number of Axial Attention Heads | 6 |
| Number of Transformer Blocks | 12 |
| Epochs | 200 |

The PDE-Refiner model was adapted from PDE-Refiner to work with our UNet model. Note that we train this model for 200 epochs while UNet and FNO are only trained for 50. See Table 12 for details.

Table 12: PDE-Refiner parameters.

| Parameter | Value |
|---|---|
| Model Size (# parameters) | 12,378,241 |
| Number of Refinement/Denoising Steps | 3 |
| Minimum Noise Scale | 4e-7 |
| Hidden Channels | 64 |
| Activation Function | GELU |
| Channel Multipliers | [1, 2, 2] |
| Num Residual Blocks Per Channel | 2 |
| Sinusoidal Time Embedding | Yes |
| Learning Rate | 1e-4 |
| Optimizer | Adam |
| Loss Function | MSE |
| Epochs | 200 |

