# OpenReview forum: "Model-Agnostic Knowledge Guided Correction for Improved Neural Surrogate Rollout"
_ICLR.cc/2025/Conference — ICLR 2025 Poster_

### Official Review · Reviewer_JFUH · 2024-11-02

**Soundness:** 2
**Presentation:** 2
**Contribution:** 2
**Rating:** 3
**Confidence:** 4

**Summary:**

The authors propose the Hybrid PDE Predictor (HyPER), which invokes costly computational simulators as knowledge-guided correction to reduce the rollout prediction errors of neural network surrogates whenever required. The proposed model relies on a reinforcement learning policy to invoke the simulator in a cost-aware manner. The resulting framework reduces the rollout error on in-distribution, out-of-distribution, and noisy data and outperforms the compared neural surrogates, at least in the studies presented by the authors.

**Strengths:**

The paper is written clearly and well-presented, and sufficient illustrations in terms of figures and tables are provided to support the claims of the authors. The hybrid concept of the rollout error correction using a simulator without needing the simulator to be differentiable is original.

**Weaknesses:**

1. Except for the switching mechanism, the proposed HyPER provides no significant contribution to the existing literature.
2. While it is true that neural surrogates produce high rollout errors at long prediction horizons, many recent works [1-6] have been carried out to address this issue. However, these are not mentioned by the authors. In order to correctly acknowledge the effectiveness of the proposed framework, a comparison against some of these frameworks is necessary.
3. The proposed framework uses a hybrid mixture of neural surrogates and costly computational simulators. However, the comparisons are performed against data-driven surrogates. Given the literature on differential physics and other hybrid methods (mentioned by the authors in the paper), it is necessary to compare the HyPER against some of the robust hybrid simulators.
4. In addition, when the neural surrogates lose temporal correlations with the initial time steps in long prediction horizons, it may be required to perform repeated predictions using the costly computational solvers. In such cases, the cost of simulation is equivalent to directly solving computational solvers like FEM and FDM.


[1] Fatone, Federico, Stefania Fresca, and Andrea Manzoni. "Long-time prediction of nonlinear parametrized dynamical systems by deep learning-based reduced order models." arXiv preprint arXiv:2201.10215 (2022).
[2] Wang, Sifan, and Paris Perdikaris. "Long-time integration of parametric evolution equations with physics-informed deeponets." Journal of Computational Physics 475 (2023): 111855.
[3] Zeng, Ailing, et al. "Are transformers effective for time series forecasting?." Proceedings of the AAAI conference on artificial intelligence. Vol. 37. No. 9. 2023.
[4] Navaneeth, N., and Souvik Chakraborty. "Waveformer for modeling dynamical systems." Mechanical Systems and Signal Processing 211 (2024): 111253.
[5] Lippe, Phillip, et al. "Pde-refiner: Achieving accurate long rollouts with neural pde solvers." Advances in Neural Information Processing Systems 36 (2024).
[6] Liu, Xin-Yang, et al. "Multi-resolution partial differential equations preserved learning framework for spatiotemporal dynamics." Communications Physics 7.1 (2024): 31.

**Questions:**

1. l25. Define RL at its first appearance.
2. Eq. (3). How are 'Error' and 'Cost' defined? The error is estimated with respect to which quantity?
3. Eq. (4). The total reward R seems to be a function of the true solution field u(x,t). Since the ground truth is not available in the inference period, how the reward will be calculated is unclear to me.
4. l199. The diffusion coefficient is taken as 0.01. Is it giving rise to laminar flow? What is the Reynolds number? It will be interesting to see the performance in high Reynolds numbers or in small diffusion coefficients. Since high rollout error generally occurs at long prediction horizons for turbulent flows more than laminar flows.
5. In the 2D Navier Stokes example, the authors consider only 20 timesteps, which is very small when considering long-term predictions. However, in the Subsurface Flow example, the authors seem to consider 100 timesteps, which is considerable.
6. How many time steps are used for training and how many for testing is not mentioned. If all the time steps are used during training, it defeats the purpose since, in practice, the neural surrogates can not be trained for finitely very long prediction horizons.
7. Table 1. Since the HyPER is pre-trained on 400 samples and the intelligent RL policy is fine-tuned on another 400 samples, the compared methods should also be trained on 800 samples since 800 datasets are already available. This seems to be acknowledged by the authors in l340.
8. Why do the fine-tuned models in Fig. 4(b) provide a higher error? Should the fine-tuned models not provide better accuracy than the pre-trained models?
9. l315. To keep a fair comparison, like the UNet and FNO are not re-trained with the changed boundary condition, the performance of HyPER should also be tested without fine-tuning the intelligent RL policy.
10. The authors should also mention the number of parameters of the models.
11. Fig. 5(a). Are the time for all the methods computed on the same type of device, i.e., CPU or GPU?
12. Alongside the time in Fig. 5(a), it will be interesting to see when the costly computational simulator is activated during inference.

---

### Official Review · Reviewer_t7Zu · 2024-11-03

**Soundness:** 1
**Presentation:** 3
**Contribution:** 1
**Rating:** 3
**Confidence:** 5

**Summary:**

The authors propose the Hybrid PDE Predictor with RL (HyPER) model,  which utilizes the reinforcement learning that combines a neural surrogate and a physics simulator to reduce surrogate rollout error significantly. This method is knowledge guided and model-agnostic. Here RL is used to decide incorporation of simulators in the loop. HyPER is compared to FNO and U-Net approaches in both accuracy and efficiency.

**Strengths:**

The paper is well-written and easy to follow. The aspect of using RL in rollout error reduction is new.

**Weaknesses:**

- The motivation and necessity of using RL with action space {0 = call surrogate, 1 = call simulator} is questionable. If physics knowledge is known, why not directly use simulators for all steps? According to Fig. 6, the computational cost reduction is not that significant. There is no accuracy comparison between HyPER and Sim-Only, but it can be predicted that Sim-Only can be more accurate. I would suggest the authors include a Sim-Only baseline in their comparisons. In Table 4, when there is no noise, the Random Policy and HyPER have almost the same accuracy, which suggests the RL here  is not meaningful.

- HyPER is compared to two surrogate baselines: UNet-Only and FNO-Only, and improves the performance significantly. However,  HyPER is knowledge-guided with PDE form known and invoked simulator, but the baselines do not require PDE knowledge. HyPER can perform better because of the knowledge imposed. The comparison is not fair.

- When changing physics conditions, the HyPER is trained with a simulator that is “fully aware of the changed boundary condition”, but “both surrogate models (UNet and FNO) are not re-trained with the changed boundary condition”. Again, the comparison is not fair.  The improvement is from the knowledge of PDE conditions.


- Could the authors clarify what “Error” and “Cost” specifically represent in formula (3)?


- The paper does not sufficiently detail the parameters and training strategies employed. The SUG and S are not specified.

**Questions:**

- What is the reason for comparing a knowledge-guided method (with physics or changing BC known) to two data-driven approaches?

- How is HyPER compared to Hybrid approaches and Sim-only approaches?

---

### Official Review · Reviewer_QQgX · 2024-11-04

**Soundness:** 2
**Presentation:** 2
**Contribution:** 2
**Rating:** 6
**Confidence:** 3

**Summary:**

This article represents an effort towards alleviating the rollout errors in neural surrogates for modeling transient dynamics. The authors assume that end-to-end training is not possible because the simulator is not differentiable. They optimize a RL policy that decides to step forward in time either with an accurate non-differentiable solver or with a neural surrogate, the resulting method is called Hybrid PDE Predictor with RL (HyPER). The reward function is a combination of an error term and a computational cost term, which limits the number of calls to the solver. Most of the article contains numerical experiments on 2D Navier Stokes and Subsurface flow applications. The experiments try to assess the benefits of the method 1) against surrogate model only approaches (UNet and FNO), 2) against change of physical conditions, 3) against noisy data, 4) against a random policy, 5) against cost/accuracy trade-offs.

**Strengths:**

This article focuses on the hard goal of reducing rollout error.
The article contains a pragmatic approach to cases where the simulation capabilities may not be differentiable (due, for example, to legacy code).
The article lays out clearly the hypothesis that the authors want to test about HyPER.
The article has an interesting way to incorporate computational cost into the training objective function.

**Weaknesses:**

The article lacks details on the model. The fact that the approach is model-agnostic does not mean that the details to make the methodology reproducible should be omitted.
- The model equations that would predict one rollout at described in Figure 2 are missing.
- It is not clear why the authors are limiting their approach to one-step auto-regresssive models instead of unrolled networks for the model or the baseline.
- Details on the computational cost of training the policy as well as its implementation are missing, which makes the results hard to reproduce. RL is known to be unstable when training, the authors should communicate the overall computational cost of training, what hyperparameters they needed to choose, how they choose them.
- The reported results don't have any error bars.
- Some terms are not explained, such as y in equation 6.
- The notations are not consistent across equations (between Eq. 4 and Eq. 6 for example).
- The illustrative 2D examples are weak baselines (see https://arxiv.org/abs/2407.07218 for more context) that are not representative of the scale of computation where such method would useful.
- There is no discussion on the convergence with respect to the number training points, and how this would scale with more challenging 3D problems.
- To finish, the authors' interpretation that the simulation step would correct the accumulated rollout error from the surrogate model is not substantiated. Such a statement is not supported by the equations because u(x,t) is unaltered to compute u(x, t+delta_t), so the error that is already accumulated in u(x,t) can't be reduced.

Note that the number of pages of the manuscript is one page over the strict maximum of ICLR submissions.

**Questions:**

Could you please share the mathematical equations for your model (see corresponding weakness above)?
Could you please share training details of HyPER (see corresponding weakness above)?
Could you explain how HyPER can theoretically reduce the already accumulate error by using a simulation?

---

### Official Review · Reviewer_fmMp · 2024-11-04

**Soundness:** 3
**Presentation:** 3
**Contribution:** 3
**Rating:** 8
**Confidence:** 3

**Summary:**

In this paper the authors tackle model prediction mismatch due to rollout, by proposing a technique that merges hybrid modeling with neural surrogates. The framework differs from previous literature given that it does not use an "only-surrogate" approach, but it also uses on-demand data from a rigorous simulator, when it identifies that this is needed. Results were presented using a 2D Navier-Stokes problem, showing that the proposed method provides improved predictions when compared to a random approach, a purely-surrogate approach, and under challenging scenarios of noise and varying physical conditions.

**Strengths:**

- The paper is well-written and organized.
- The paper merges concepts from different fields (hybrid modeling + neural surrogates) in interesting ways.
- The results that are presented are favorable for the proposed method.

**Weaknesses:**

- In the introduction, and in the results presented,  motivation describing real scenarios of rollout is missing. The authors mention the fact that rollout is an issue for simulations, but do not cite or list any real examples. Providing such examples and refs even in the introduction would strengthen the paper.
- The authors present results only using a 2D Navier-Stokes benchmark problem. This further points to the previous comment on motivation. Moreover, using only this problem does not address the scalability of the proposed approach. How would this approach perform in terms of accuracy and computational cost for larger simulations with many dimensions, parameters, variables affecting predictions? If the authors cannot a larger example in supplementary, they could at least add a discussion on this.
- There is a large body of literature in hybrid modeling (starting from the 1990's, where different structures of model correction or different fidelity of models are embedded) that is relevant to this work that is not mentioned at all in this paper. The authors should include an earlier reference and clearly describe the novelty of the proposed work compared to earlier work as well.
- The comparisons presented with only pre-trained surrogates do not seem as fair, unless I have misunderstood the approach. The HyPER approach continuously updates the models by getting new data from high-fidelity simulation. The only-surrogate approaches do not (again, unless I misunderstood). It is thus expected that the HyPER approach would outperform all else. This can be ok, if one considers that the novelty of the HyPER framework is it's adaptive nature. However, given that when new data comes, some re-training happens, would it not be fair to allow for the surrogate-only approaches to also be re-trained with new data? It is likely that they would still perform worse, or require more training time, but such a comparison would help better explain the true novelty of the framework.

**Questions:**

- What are some real science or engineering-based problems/case studies that suffer from rollout errors? What is their dimensionality and what time-scales are relevant for these problems with respect to decision-making (e.g., control problems where one needs to perform an action within fractions of second or other?).
- Based on above answer, how would this method scale to real systems (if they are different to the presented 2D N-S problem)?
- How is this work relevant or different to earlier hybrid modeling work developed in process systems engineering or for control starting in the 1990's?
- How would only surrogate techniques predictive errors change if they were re-trained with new data from simulator (if this was not done already)? In other words, is it the hybrid modeling structure or the adaptability or both that are novel and effective in this work?

---

### Meta-Review · Area_Chair_7ux1 · 2024-12-23

**Metareview:**

Thank you for your submission to ICLR. This paper proposes the Hybrid PDE Predictor with Reinforcement Learning (HyPER) method for neural surrogate rollout, with the goal of reducing surrogate rollout error. HyPER combines a neural surrogate model, the physics simulator, and a reinforcement learning model, to reduce costs (e.g., over the pure-simulator), while achieving less approximation error than competing methods (e.g., compared to neural-surrogate-only methods).

There were some concerns from reviewers about the use of methodology involving hybrid simulators, the inclusion of additional baseline comparisons, and the emphasis on certain metrics and timing results in experiments. However, I feel that the authors addressed these comments sufficiently in their rebuttal process: they justified the value of RL/simulator hybrid methods, they compared against a thorough set of baselines, and explained or added requested metrics and timing results. I also feel that hybrid methods such as this can have value in practice to the community, even if they do come with computational cost tradeoffs.

**Additional Comments On Reviewer Discussion:**

After rebuttal and discussion, reviewer fmMP and QQgX felt that all of their comments were addressed, and both raised their scores. Reviewer t7Zu had a healthy discussion with the authors but remained unconvinced (though did adjust scores as well). However, the authors gave their arguments and responses to all questions posed by reviewer t7Zu, which I feel were sufficient.

---

### Decision · Program_Chairs · 2025-01-22

Accept (Poster)